# Assessing the Effectiveness of Eslicarbazepine Acetate in Reducing Audiogenic Reflex Seizures in the GASH/Sal Model of Epilepsy

**DOI:** 10.3390/biomedicines12051121

**Published:** 2024-05-18

**Authors:** Jaime Gonçalves-Sánchez, Thomas Ramírez-Santos, Dolores E. López, Jesús M. Gonçalves-Estella, Consuelo Sancho

**Affiliations:** 1Department of Cell Biology and Pathology, School of Medicine, University of Salamanca, 37007 Salamanca, Spain; 2Institute for Biomedical Research of Salamanca (IBSAL), 37007 Salamanca, Spain; 3Institute of Neuroscience of Castilla y León, 37007 Salamanca, Spain; 4Department of Surgery, School of Medicine, University of Salamanca, 37007 Salamanca, Spain; 5Department of Physiology and Pharmacology, School of Medicine, University of Salamanca, 37007 Salamanca, Spain

**Keywords:** animal models of epilepsy, anticonvulsants, epilepsy, reflex, eslicarbazepine acetate, seizures

## Abstract

Eslicarbazepine acetate (ESL) is a third-generation antiepileptic drug indicated as monotherapy for adults with newly diagnosed epilepsy and as adjunctive therapy for the treatment of partial seizures. Our aim was to assess the effectiveness and safety of both acute and repeated ESL administration against reflex audiogenic seizures, as shown by the Genetic Audiogenic Seizures Hamster from Salamanca (GASH/Sal). Animals were subject to the intraperitoneal administration of ESL, applying doses of 100, 150 and 200 mg/kg for the acute study, whereas a daily dose of 100 mg/kg was selected for the subchronic study, which lasted 14 days. In both studies, the anticonvulsant effect of the therapy was evaluated using neuroethological methods. To assess the safety of the treatment, behavioral tests were performed, hematological and biochemical liver profiles were obtained, and body weight was monitored. In addition, the ESL levels in blood were measured after the acute administration of a 200 mg/kg dose. Treatment with ESL caused a reduction in seizure severity. No statistically significant differences were detected between the selected doses or between the acute or repeated administration of the drug. To summarize, the intraperitoneal administration of ESL is safe and shows an anticonvulsant effect in the GASH/Sal.

## 1. Introduction

According to the International League Against Epilepsy (ILAE), “epilepsy is a neurological disease defined by the presence of two or more unprovoked seizures occurring more than 24 h apart, or one unprovoked seizure with a high risk of recurrence” [1,2]. It is a very common condition, since it is estimated that 10% of the general population will experience at least one seizure throughout their life, and that between 1 and 2% will develop epilepsy [3].

Although antiepileptic drugs (AEDs) are the first-line treatment for epilepsy, in about a third of patients, effective seizure control is not achieved with monotherapy or with the combination of several AEDs, which is known as refractory epilepsy [4,5]. Furthermore, in recent decades, plenty of new generation AEDs have been discovered [6]. These medications show advantages in tolerance, drug interactions and their therapeutic spectrum, but they are not always more effective in controlling seizures [7,8]. Therefore, there is still a need to search for new medications to improve seizure control, with the aim of increasing the proportion of those affected by refractory epilepsy who achieve the effective management of the disease [9].

One of these new-generation AEDs is eslicarbazepine acetate (ESL) [10], which was approved in 2009 by the European Medicines Agency (EMA) (Zebinix™, Laboratorios Bial, Madrid, Spain) and in 2013 by the US Food and Drug Administration (FDA) (Aptiom™, Marlborough, MA, USA) as a once-a-day adjunctive therapy in adults with refractory partial-onset seizures, with or without secondary generalization [11]. Since then, its indication has been expanded to monotherapy for these epilepsies and as adjunctive therapy for children over 6 years of age in Europe, and both as monotherapy and adjunctive therapy in the USA in children over 4 years of age [12].

Research supports ESL’s effectiveness, showing a significant reduction in seizure frequency, as well as favorable safety, with mild to moderate adverse reactions, such as dizziness, headache and drowsiness, mostly during the first weeks of treatment [13,14,15,16].

ESL is a dibenzoazepine structurally related to carbamazepine and oxcarbazepine. However, these drugs show differences in their effectiveness, pharmacokinetics, pharmacodynamics and tolerability [17]. Recent studies have indicated that treatment with two dibenzoazepines can be more effective that treatment with only one, and transition from a dibenzoazepine agent to another (specially to ESL) can be beneficial for some patients [18]. On the one hand, unlike carbamazepine, ESL is not metabolized to epoxide, which minimizes enzyme induction, and thus ESL would provide a safer profile and fewer drug–drug interactions [19,20]. On the other hand, the main difference between oxcarbazepine and ESL is that the latter is stereoselectively metabolized to S-licarbazepine (eslicarbazepine), whereas the breakdown products of oxcarbazepine include a higher amount of R-licarbazepine; this is also an active metabolite, but it is believed to be more toxic and less effective [21]. Thus, the anticonvulsant effect of ESL is due mainly to its active metabolite eslicarbazepine, which is thought to stabilize the inactive state of voltage-gated sodium channels, preventing repetitive neuronal discharges during an epileptic seizure [22,23,24,25].

In this study, we aim to describe the effects of acute and repeated ESL treatment in a convulsive seizures model, the GASH/Sal hamster (Genetic Audiogenic Seizures Hamster from Salamanca). The GASH/Sal shows reflex tonic–clonic audiogenic seizures of troencoencephalic origin in response to an intense sound stimulus [26]. Recently, among other research lines, several treatments for epilepsy have already been studied using this model [27,28,29,30,31]. In addition, although the effects of ESL have been tested in animal models [10,32,33,34], ESL’s effects have not been described in any animal model of audiogenic seizures. Even if audiogenic seizures and reflex epilepsies in general are a rather small representation of the global seizures in humans, they are present in the population with a prevalence rate of 4–7% in all epilepsy patients [35,36]. But, more importantly, models of reflex seizures show the advantage of being able to induce seizures when desired, and thus the effectiveness of drugs can be evaluated easily, avoiding interactions with other chemicals [37]. The peculiarities of this epilepsy model, as seizures are always provoked with the same type of stimulus and have the same characteristics, allow greater homogeneity in the results compared to the variability that exists in models that present spontaneous seizures.

Therefore, we believe that the proposed study contributes to expanding knowledge of the drug and its potential applications.

## 2. Materials and Methods

### 2.1. Animals

A total of 12 GASH/Sal hamsters, both males and females aged 2–6 months, were used in this study. They were obtained from the inbred strain maintained in the Animal Facility at the University of Salamanca.

### 2.2. Ethics Statement

The experimental and animal handling procedures followed established protocols in Directive 2010/63/EU of the European Parliament and Council, in the current Spanish legislation (Royal Decree 1201/05), and in accordance with those established by the Institutional Bioethics Committee (approval number 380). Throughout the work, we employed procedures to minimize pain and improve the well-being of the animals used from birth to death. Also, we used the smallest number of animals required to obtain sufficient data to answer the questions posed. Reducing the number of animals used was achieved in two ways. Firstly, the experimental design allowed each animal to be its own control. Secondly, the same animals from the acute experiments were also used in the subchronic study and for the determination of the drug levels in blood.

### 2.3. Experimental Desgin

Both an open-field test and a seizure recording protocol were performed on all animals when serving as a control. Subsequently, a 200 mg/kg dose of ESL was administered intraperitoneally (i.p.) (Zebinix oral suspension, Laboratorios Bial, Madrid, Spain). Sixty minutes after the drug injection, subjects were exposed to acoustic stimulation to observe the possible effects on the seizure severity. This experiment was repeated, in the same specimens, after a recovery period of four days between tests, with doses of 100 mg/kg and 150 mg/kg of ESL. Subsequently, the animals underwent a recovery period of seven days, and the subjects were randomly divided into two groups. One of the groups (*n* = 6) began to receive prolonged treatment with ESL at a daily dose of 100 mg/kg for 14 days, in which seizures were evaluated after 7, 10, and 14 days of treatment. In addition, behavioral tests were performed in this group after days 6 and 13 of treatment. Furthermore, body weight was monitored. Finally, blood samples were collected at the end of the experiment for hemogram analysis and the measurement of biochemical parameters. The specimens from the second group (*n* = 6) received an acute dose of 150 mg/kg to study the acute effect of the drug on general behavior using open-field tests. After several days of recovery, blood samples were collected at 15, 30, 45 and 60 min after the injection of ESL (200 mg/kg) to elicit the concentration of the active ingredient in blood. At the end of the experiment, animals were euthanized under deep anesthesia by the inhalation of CO_2_.

### 2.4. Acoustic Stimuli

The procedure was performed according to the laboratory protocol [26]. The hamsters were placed in a cylindrical arena (height = 50 cm, diameter = 37 cm) and allowed to acclimatize for one minute. GASH/Sal hamsters were then exposed to white noise (0–18 kHz; 115–120 dB) to induce audiogenic seizures. Exposure continued until they showed tonic–clonic convulsions, or until a minute had elapsed, whichever occurred first. When the subjects received treatment with ESL, the test was performed 60 min after drug administration. This procedure was video-recorded for further neuroethological evaluation (see below).

### 2.5. Neuroethological Analysis

The anticonvulsant effect of the therapy was evaluated using the categorized seizure severity index (cSI) (Table 1). This index has enabled researchers to use statistical methods for the assessment of seizure severity in rodents [38]. In addition, the latency of the wild-running phase was also noted.

For each session, the characteristic progression of the motor seizure and kinetic postural components, as well as their duration, were determined based on our own observations. Video recordings began one minute prior to sound exposure and continued until the animal recovered from the stupor.

Ethomatic software [39] was used to analyze the characteristics of the seizures recorded on video. Three 1 min windows corresponding to the animals’ behavior before, during, and after sound exposure were analyzed [39]. In the analysis, carried out with Microsoft PowerPoint 365, the animals that had seizures after exposure to sound (cSI ≠ 0) were separated from those who did not have them (cSI = 0). The diagram shows the behavioral sequences described second by second as a probabilistic chain of events, evaluating the interdependence in dyadic interactions (pairs of behaviors). Figure 1 contains the dictionary of behaviors used and the main characteristics of the Ethomatic software.

### 2.6. Open-Field Tests

This procedure was performed before treatment, after the acute administration of ESL (150 mg/kg), and at days 6 and 13 of repeated treatment (100 mg/kg). In all cases, the tests were carried out 60 min after intraperitoneal injection and at the same time of day to avoid variabilities associated with the circadian rhythm. The GASH/Sal hamsters were placed in the center of an 80 cm diameter arena rounded by a 30 cm high wall, which was divided into three concentric zones, each representing one third of the surface. The tests lasted 12 min and were videorecorded from above. ANY-MAZE software (Stoelting Co., Wood Dale, IL, USA, v 6.16) was used for data collection, including the distance travelled, time spent in the central zone, and both grooming and rearing duration.

### 2.7. Hematological and Biochemical Liver Profiles

During euthanasia, we collected blood through cardiac puncture in EDTA-containing tubes and evaluated various hematological parameters using a ADVIA 120 cytometer (Bayer, Leverkusen, Germany).

For biochemical analysis, we collected 400 µL of blood in tubes without anticoagulants, evaluating various parameters of the hepatic metabolism, such as albumin, aspartate aminotransferase, alanine aminotransferase, bilirubin and total protein, using the Spotchem II Liver-1 kit, #33925, Menarini Diagnostic, Badalona, Spain.

### 2.8. Analysis of Drug Concentration in Blood

In this stage, 150 µL of blood was drawn from the cranial vena cava of the specimens treated with ESL (200 mg/kg) [40]. These extractions were performed 15, 30, 45 and 60 min after drug injection. The procedure was executed under inhalation anesthesia with sevoflurane, and the animals were rehydrated with an equivalent amount of saline. After clot formation, the samples were centrifuged for 10 min at 10,000 rpm to obtain serum. The serum samples were analyzed using HPLC/MS, performed by the Elemental Analysis, Chromatography and Mass Service of the University of Salamanca, to obtain the corresponding concentration curve.

### 2.9. Statistical Analysis

Data analysis was performed using IBM SPSS Statistics software, v.26 (SPSS Inc., Chicago, IL, USA). The statistical significance was *p* < 0.05 (*), *p* < 0.01 (**), and *p* < 0.001 (***). All data are shown as mean ± standard deviation (SD).

Repeated measures tests for non-parametric data (Friedman test, Durbin–Conover for post hoc comparisons) were used to evaluate the seizure severity before and after acute and subchronic treatment with ESL. To compare the wild running latencies, since some animals in the treated groups did not show wild running and these data did not follow a normal distribution, non-parametric tests were used (Kruskall–Wallis and Dwass–Steel–Critchlow–Fligner for comparisons).

The flowcharts were created using the Ethomatic program. Each behavioral sequence was represented graphically with a line proportional to the statistical result of the χ^2^ test, with the width of the arrows shown proportional to the logarithmic values of this parameter.

To compare the results of the open-field tests, after checking the normality (Shapiro–Wilk) and homoscedasticity of the data (Levene’s test), one-way ANOVA tests were used (Tukey’s test for post hoc comparisons).

## 3. Results

### 3.1. Effects of ESL on Seizures

In the first record, which was performed before treatment to serve as a control, the GASH/Sal hamsters showed high seizure severity index scores, which is usual in the experimental model (cSI = 6.75 ± 1.95). Acute treatment with all the selected doses of ESL caused a statistically significant reduction in seizure severity (cSI = 1.64 ± 1.57 for 100 mg/kg; cSI = 1.55 ± 2.54 for 150 and 200 mg/kg; *p* < 0.001 for each dose). Furthermore, no differences were detected regarding the anticonvulsant effect between the selected doses (Figure 2A). Acute treatment with ESL was also associated with a significant increase in the wild running latency, for all doses of ESL (*p* < 0.05) (Figure 2B).

In addition, the repeated administration of ESL at doses of 100 mg/kg did not produce a more potent anticonvulsant effect, compared to the acute administration of the same dose (Figure 3A). Furthermore, no changes were observed in the wild running latency between the acute and subchronic experiment (Figure 3B).

As can be seen in the results of the neuroethological analysis, the untreated animals showed the typical features of the seizures of the GASH/Sal (Figure 4). Before exposure to sound, clusters of exploratory behaviors (SN, WA, ER, SC, TR, SC) and grooming (GRF, GRH, LIC) were observed, both shown in blue. When animals were exposed to sound, a startle behavior (STA) immediately occurred, followed by a wild running phase (RU, GL, GR, JP, AF), shown in yellow. In most GASH/Sal, the sound stopped when the animals showed tonic seizures (TCV), which were followed by head ventral flexion (HFL), limb hyperextension (HP1, HP2), and clonic seizures (CCV1, CCV2, CVR2, CCVg). These seizure behaviors are shown in red in Figure 4. Finally, during the postictal period, the animals presented immobility or stupor (PIM), which could be associated with myoclonus and respiratory alterations (MYO, TCP, DYS; in orange) or orofacial automatisms (GRF, MT; in purple). After a few minutes, the animals normally recovered from stupor and displayed exploratory and grooming behaviors again.

As stated above, acute treatment with ESL had a clear anticonvulsant effect. Moreover, there was a complete absence of seizures (cSI = 0) in a high ratio of animals treated with a single dose of ESL (Figure 5, left column). Startle behavior (STA) was observed in these rodents after sound exposure, but this did not trigger wild running. Furthermore, an increase in grooming behaviors was observed in these animals. Nevertheless, there was another group of subjects that, even after receiving acute pharmacological treatment, showed seizures (cSI ≠ 0) (Figure 5, right column). In this group, there was a decrease in the association between tonic–clonic, post-ictal and orofacial automatisms; because of the effect of the therapy, most animals did not display the complete behavioral sequence present in the typical seizures seen in the GASH/Sal model.

The subchronic treatment with ESL (100 mg/kg) showed an anticonvulsant effect similar to the one obtained in the acute experiment. There were no statistical differences in terms of the anticonvulsant effect observed between the tests performed on days 7, 10 and 14, even though the percentage of animals that showed a total response, defined as an absence of seizures, was variable between trials (Figure 6, left column). The total absence of tonic–clonic behaviors on day 10 of treatment is remarkable. In addition, the heterogeneity of the results obtained on day 14 of subchronic treatment with ESL stands out, given that there was one animal that showed wild running, but this was not followed by stupor, which is why exploratory and grooming behaviors are shown. In contrast, there was another animal with cSI = 0 that did not show wild running but did display stupor and tachypnea behaviors. Finally, it is worth noting that for doses of 100 mg/kg, no orofacial automatisms (in purple) were observed in any trial, neither in the acute nor in the subchronic experiment. 

### 3.2. Effects of ESL on Behavior

A statistically significant reduction in the rearing time was observed in the groups treated with ESL, both after a single dose of 150 mg/kg (*p* < 0.05) and in the second week of subchronic treatment with 100 mg/kg (*p* < 0.01). For the rest of the parameters studied, no statistically significant differences were detected, and high variability was observed between trials (Figure 7).

### 3.3. Evaluation of Overall State

The hematological and biochemical parameters obtained were considered within the theoretical reference values for the hamster [41], except for the aspartate transaminase values, which were slightly lower than normal (Figure 8A). Furthermore, there were differences in the alanine transaminase level compared to untreated animals, based on the data from previous research [29]. On the other hand, the repeated administration of the drug did not alter the body weight of the animals (Figure 8B).

### 3.4. ESL Levels in Blood

The concentration of ESL in the blood gradually increased in the first sixty minutes after drug administration (200 mg/kg), from 50 ± 13.93 ng/mL, fifteen minutes after administration, to 89.48 ± 22.45 ng/mL an hour later. Thirty and forty-five minutes after injection, values of 55.95 ± 14.82 ng/mL and 77.33 ± 17.76 ng/mL were obtained, respectively.

## 4. Discussion

Both the acute and repeated administration of ESL had a clear anticonvulsant effect on GASH/Sal, without significant differences between the selected doses. The method of neuroethological analysis described here, which was used to evaluate seizures, has been validated in the GASH/Sal [26], in other models of epilepsy [39] and in patients [42]. The anticonvulsant effect of ESL has been previously demonstrated in several animal models, among which the mouse maximal electroshock (MES) [25,43], the 6 Hz psychomotor [23], amygdala kindling [33], pilocarpine [44] and genetic models [45] stand out. However, the study is novel in the sense that the effect of ESL has been demonstrated for the first time in a model of audiogenic seizures, which contributes to knowledge of the drug. Moreover, the GASH/Sal model has been validated as a tool to assess the efficacy and safety of pharmacological treatments, since the anticonvulsant effect of both classic and more recently implemented AEDs, such as ESL, has been corroborated [27,28,29]. Finally, we highlight the main advantage of the model, which allows seizure induction on demand without the use of other pharmacological agents that could interfere with the AEDs studied. Moreover, ESL has proven to be safe with the selected doses, despite causing slight alterations in transaminase levels. In previous research by our group, the combination of cannabidiol and valproic acid caused a similar effect [29]. In animal studies, nephrotoxicity has been observed in rats, but not in mice or dogs, in addition to hepatic centrilobular hypertrophy with repeated doses in mice and rats, consistent with an induction of hepatic microsomal enzymes [46]. Recently, it has been demonstrated that the ESL metabolism differs across species. Mice, hamsters, and rabbits show a more similar ESL metabolism to humans than dogs, rats, or monkeys, since most ESL is metabolized to eslicarbazepine, and no ESL is found after oral administration [47]. However, it is important to note that our study has certain limitations. Firstly, we did not collect liver samples at the end of the experiment, and therefore it was not possible to perform histopathological evaluations. Secondly, there is the possibility that there are hepatic lesions that do not have an effect on the plasmatic levels of enzymes. However, the pharmacokinetics of eslicarbazepine are not affected significantly due to mild to moderate liver failure [48]. In addition, including biochemical profiles and histological evaluations of the kidney, would also provide a more comprehensive safety assessment. In the clinic, AED liver injury is thought to have significantly decreased over the last two decades with the introduction of new medications [49]. Nonetheless, although considered safe, there have been very rare, isolated reports of clinically apparent liver injury or increased alanine aminotransferase levels associated with eslicarbazepine [50].

Furthermore, the doses selected were similar to those found in the literature. For example, in the mouse 6 Hz model of psychomotor seizures [51], lamotrigine-resistant rats [52] or kindling models [33,53], doses between 100 and 300 mg/kg showed anticonvulsant effects. In our study, although seizure freedom was not reached in all the trials, the use of a higher dose in the experimental model with the aim of achieving a greater anticonvulsant effect was discarded, since the administration of a dose of 300 mg/kg of ESL caused severe sedation, and even cannibalistic behavior. On the other hand, the acoustic stimulation of the animals was carried out 60 min after drug injection. There is a possibility that the results would have differed if the tests had been performed at a different time after the administration of ESL, especially in the acute study. In addition, ESL was thought to be metabolized at a higher rate in the hamster, since in CD-1 mice, the maximum concentration in plasma is reached after 15 min, and after 60 min in the brain [47]. This was one of the factors that led to the seizures being evaluated in a short period of time after the administration of the drug. Nevertheless, although it has been reported that the maximum concentration of the active ingredient in the blood of humans is between 2 and 3 h, its half-life is between 13 and 20 h depending on the dose [54,55,56]. Therefore, even though our results support the literature, the study could have been more complete if the blood samples had been drawn later after the administration of ESL.

Previous studies by our research group using other AEDs, intraperitoneal vehicle administration [29] or repeated acoustic stimulation with several days of separation between recordings did not find changes in the animals’ seizures [30]. Because of these data, it was possible to omit the simulated group from the experiments, reducing the number of animals used. Additionally, we have data from open-field tests in GASH/Sal hamsters treated with saline solution compared to other AEDs, so we also knew the effects of repeating the behavioral tests and intraperitoneal injection [29]. In the experiments mentioned, the repetition of the open-field test in the simulated group caused a decrease in the distance travelled, elevations and grooming. The animals treated with ESL showed similar results to the animals treated with vehicle in these studies, except regarding grooming, since ESL seems to slightly increase grooming time. However, no statistically significant differences were detected for this parameter. Nevertheless, we are aware that different doses were used in the open-field tests in the acute (150 mg/kg) and the subchronic studies (100 mg/kg). Although they had a similar effect on seizures, comparing the results from both experiments should be undertaken with caution and this difference should be kept in mind. The main purpose of the open-field tests was to rule out severe side effects on general behavior with the selected doses, and we think we achieved this goal.

## 5. Conclusions

The acute intraperitoneal administration of ESL at doses of 100, 150 and 200 mg/kg had an anticonvulsant effect on the GASH/Sal hamsters, although a total absence of seizures was not observed. No statistical differences between doses were detected regarding seizure reduction. Repeated treatment with ESL (100 mg/kg) had a similar effect on seizures. The treatment was considered safe at doses of 200 mg/kg or lower, since no adverse effects on general behavior, the hemogram or body weight were observed. However, slight alterations in the transaminase levels were detected. This is the first report in the literature in which the effects of ESL are described in a model of audiogenic seizures.

## Figures and Tables

**Figure 1 biomedicines-12-01121-f001:**
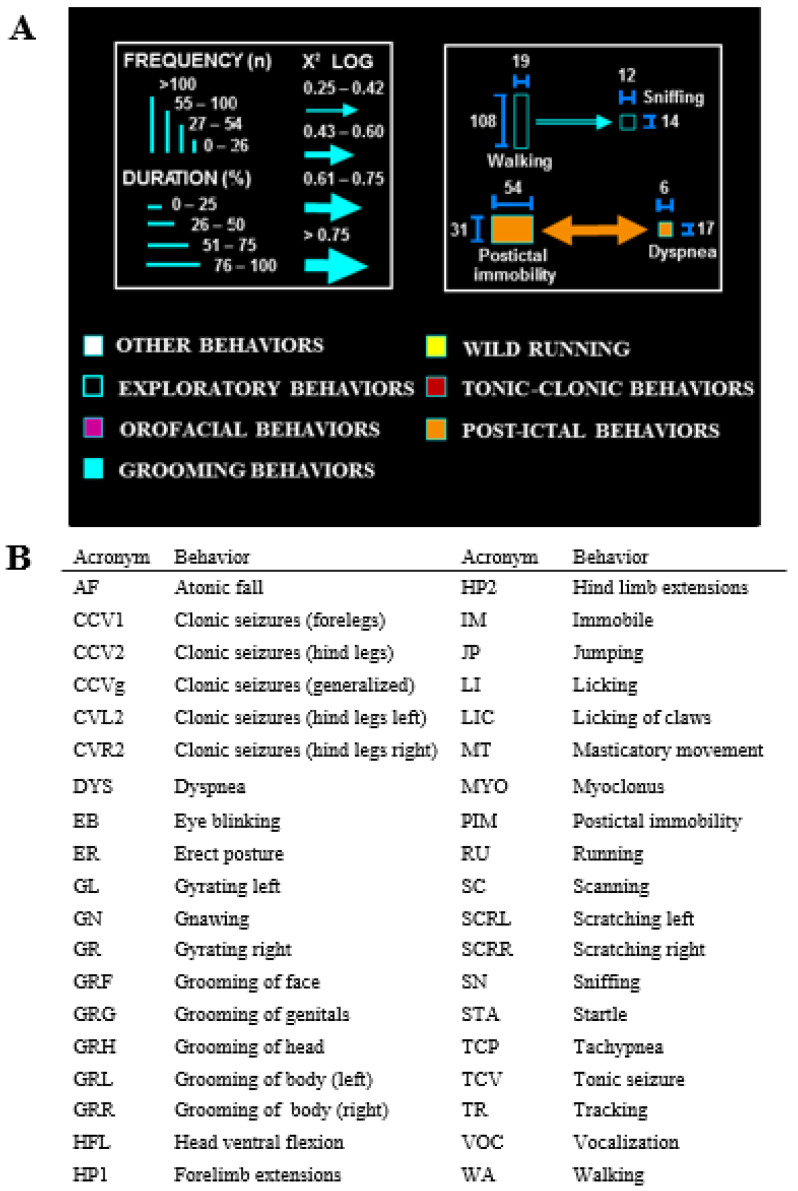
Methods for neuroethological analysis: (**A**) Ethomatic software and flowchart calibration. The program presents the frequency and duration of each behavior in the observation windows, proportional to the height and width of the rectangles that represent those behaviors. The program also performs the statistical analysis, verifying the possible significant associations between pairs of behavioral items, calculating the χ^2^ values. In this case, the direction of the arrows indicates the association between two behaviors, and their width is proportional to their χ^2^ (statistical significance). The behaviors, grouped in clusters, are identified with a specific color for each grouping. The behaviors indicated in the flow charts are collected in a dictionary. (**B**) The dictionary of behaviors used to elaborate the flowcharts.

**Figure 2 biomedicines-12-01121-f002:**
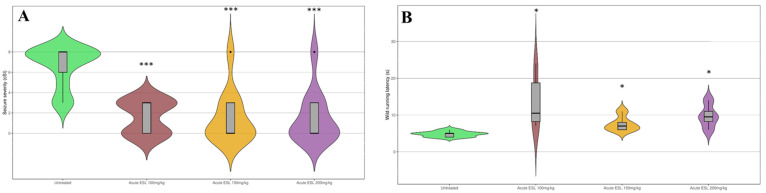
Effect of acute treatment with ESL on seizures. The violin plots, made with the R package ggplot2 3.5.0, show the performance of the experimental groups for different parameters. (**A**) Effect of acute administration of ESL on seizure severity. Doses of 100, 150 and 200 mg/kg produced a significant reduction in seizure severity (cSI index) when compared with the untreated condition. No differences were observed between doses. (**B**) Effect of acute administration of ESL on wild running latency. ESL produced an increase in this parameter, reaching statistical significance for all the doses tested. The horizontal thick line represents the median, and the boxes represent the interquartile range. The thin black line represents the rest of the distribution, except for points that are determined to be “outliers” (black dots) using a method that is a function of the interquartile range. ‘*’ (*p* < 0.05); ‘***’ (*p* < 0.001).

**Figure 3 biomedicines-12-01121-f003:**
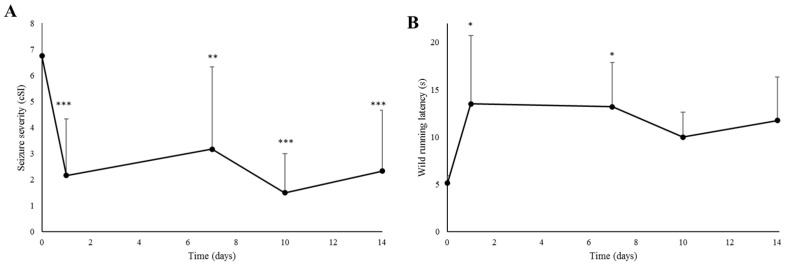
Effect of repeated treatment with ESL on seizures. (**A**) Effect of subchronic administration of ESL on seizure severity. Daily intraperitoneal doses of 100 mg/kg produced significant reductions in the cSI scores. Repetitive administration of the drug did not induce a cumulative effect. (**B**) Effect of subchronic administration of ESL on wild running latency. Results are shown as mean ± hemi standard deviation. ‘*’ (*p* < 0.05); ‘**’ (*p* < 0.01); ‘***’ (*p* < 0.001).

**Figure 4 biomedicines-12-01121-f004:**
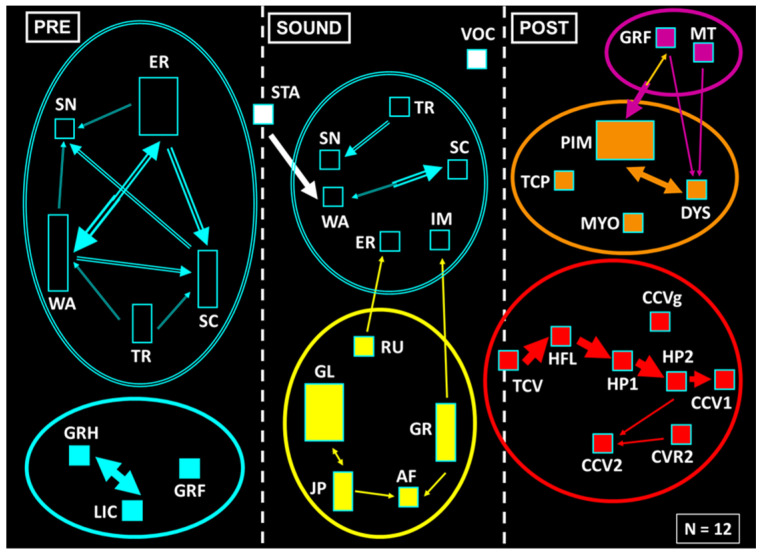
Results of the neuroethological analysis before treatment. The complete sequence of behaviors present in the typical seizures displayed by the GASH/Sal was observed. See Figure 1 for flowchart calibration and abbreviations.

**Figure 5 biomedicines-12-01121-f005:**
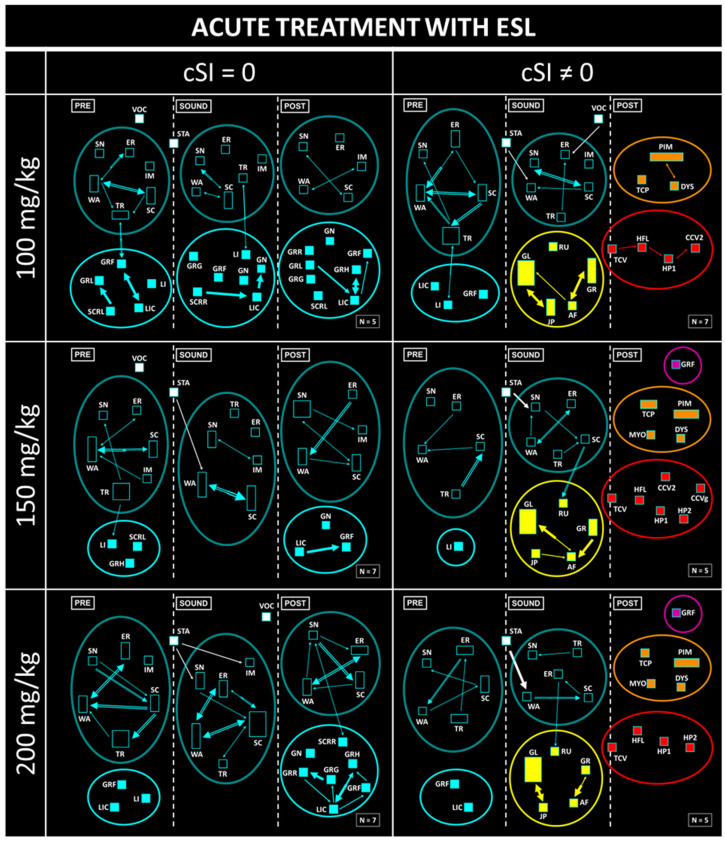
Results of the neuroethological analysis after acute administration of ESL. Note the different doses used (100, 150, and 200 mg/kg) and the classification of the subjects based on whether seizures were present (cSI ≠ 0) or not (cSI = 0).

**Figure 6 biomedicines-12-01121-f006:**
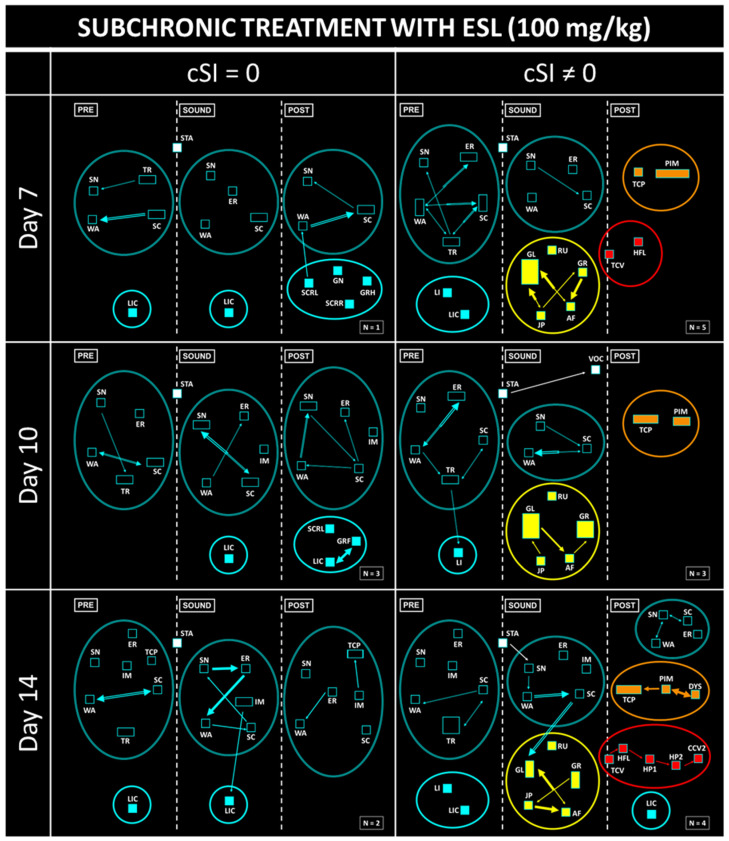
Results of the neuroethological analysis after subchronic administration of ESL (100 mg/kg). Note the evaluation of seizures at different times (at days 7, 10 and 14 of treatment) and the classification of the subjects based on whether seizures were present (cSI ≠ 0) or not (cSI = 0).

**Figure 7 biomedicines-12-01121-f007:**
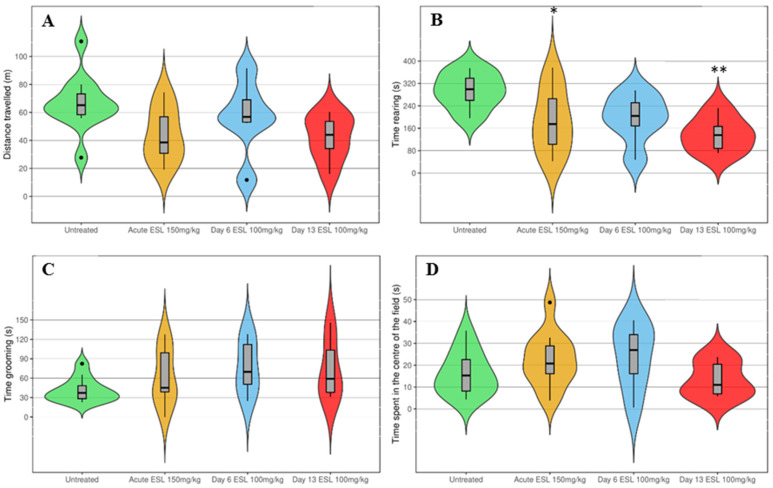
Effect of acute (150 mg/kg) and subchronic (100 mg/kg) ESL on general behavior. The violin plots, made with the R package ggplot2 3.5.0, show the open-field test scores obtained in distance travelled (**A**), time rearing (**B**), time spent in the center of the field (**C**), and grooming duration (**D**). The horizontal thick line represents the median, and the boxes represent the interquartile range. The thin black line represents the rest of the distribution, except for points that are determined to be “outliers” (black dots) using a method that is a function of the interquartile range. ‘*’ (*p* < 0.05); ‘**’ (*p* < 0.01).

**Figure 8 biomedicines-12-01121-f008:**
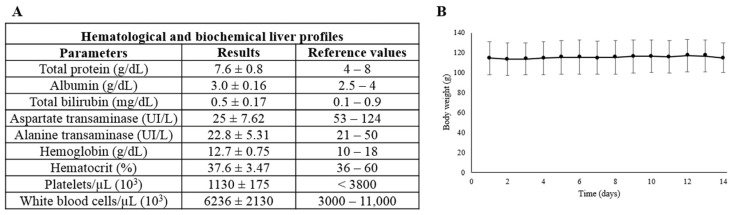
Effect of repeated (100 mg/kg) ESL on overall state. (**A**). Hematological and biochemical liver profiles. All data are shown as mean ± standard deviation. Reference values are taken from [29,41] (**B**). Effect of ESL on body weight. Treatment with ESL did not produce significant changes.

**Table 1 biomedicines-12-01121-t001:** Categorized seizure severity index (cSI).

Seizure Behaviors	cSI
No seizures	0
One wild running	1
One wild running (running plus jumping plus atonic fall)	2
Two wild runnings	3
Tonic seizure (opisthotonus)	4
Tonic seizures plus generalized clonic seizures	5
Head ventral flexion plus cSI5	6
Forelimb extension plus cSI6 ^a^	7
Hind limb extension plus cSI7 ^a^	8

^a^ Categories, which are generally followed by hind limb clonic seizures.

## Data Availability

The data presented in this study are available upon request from the corresponding author.

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
