# Peer review of "Assessing the Effectiveness of Eslicarbazepine Acetate in Reducing Audiogenic Reflex Seizures in the GASH/Sal Model of Epilepsy"

_biomedicines, 2024, doi:10.3390/biomedicines12051121_

Round 1

Reviewer 1 Report

Comments and Suggestions for Authors

The manuscript entitled "Study of the effect of eslicarbazepine acetate on a genetic model of audiogenic reflex seizures, the GASH/Sal" aims to assess the effectiveness and safety of both acute and chronic administration of eslicarbazepine acetate (ESL) in reducing reflex audiogenic seizures in the GASH/Sal model. The study involved intraperitoneal administration of ESL at different doses, and the anticonvulsant effect was evaluated using neuroethological methods. Safety assessment included behavioral tests, hematological and biochemical liver profiles, and monitoring of body weight. Treatment with ESL demonstrated a significant reduction in seizure severity, with no significant differences observed among doses or between acute and chronic administration. The findings suggest that intraperitoneal administration of ESL is safe and exhibits an anticonvulsant effect in the GASH/Sal model. However, the manuscript requires major revisions before considering it for publication.

SPECIFIC COMMENTS TO THE AUTHORS:

  1. Title: I recommend revising the title to better reflect the content of your study. A suggestion is: "Assessing the Effectiveness and Safety of Eslicarbazepine Acetate in Reducing Audiogenic Reflex Seizures in a Genetic Model (GASH/Sal)."
  2. Abstract: Please ensure that the keywords are capitalized and selected using MESH for better indexing and searchability.
  3. Introduction: To enhance the introduction, consider providing a more comprehensive background by including recent publications to ensure the information is current. Emphasize the novelty of your research to highlight its unique contribution. Additionally, in the last paragraph of the introduction, provide a clear statement of the study's aim to help readers understand the objectives and methodology.
  4. Methods and Results: To address the safety of ESL, I suggest including histopathological evaluations of the liver and kidney. Additionally, consider studying the biochemical profiles of the kidney for a more comprehensive safety assessment.
  5. Discussion: Organize the discussion section to improve clarity and coherence. Address various aspects of the study one by one and compare your results with those of other relevant studies. Updating the references in the discussion section to include recently published papers will contribute to a more comprehensive and up-to-date discussion.
  6. Conclusion: Include a concise conclusion to summarize the results of your study.

Comments on the Quality of English Language

The manuscript can be checked for some minor English errors. 

Author Response

  1. Title: I recommend revising the title to better reflect the content of your study. A suggestion is: "Assessing the Effectiveness and Safety of Eslicarbazepine Acetate in Reducing Audiogenic Reflex Seizures in a Genetic Model (GASH/Sal)."

Answer: The title has been changed to “Assessing the Effectiveness of Eslicarbazepine Acetate in Reducing Audiogenic Reflex Seizures in the GASH/Sal model of epilepsy”. We decided to remove the term safety given that the evaluation of this aspect was not very detailed.

  1. Abstract: Please ensure that the keywords are capitalized and selected using MESH for better indexing and searchability.

Answer: Keywords have now been changed to “Animal models of epilepsy; Anticonvulsants; Epilepsy, reflex; Eslicarbazepine acetate; Seizures;”, which are MeSH terms.

  1. Introduction: To enhance the introduction, consider providing a more comprehensive background by including recent publications to ensure the information is current. Emphasize the novelty of your research to highlight its unique contribution. Additionally, in the last paragraph of the introduction, provide a clear statement of the study's aim to help readers understand the objectives and methodology.

Answer: We agree with your comment and believe that the proposed changes improve the introduction section. Although the available literature is not very extensive, we have included new references. To emphasize why our research is important and what the most important objectives of the study are, we added a brief paragraph at the end of the introduction.

  1. Methods and Results: To address the safety of ESL, I suggest including histopathological evaluations of the liver and kidney. Additionally, consider studying the biochemical profiles of the kidney for a more comprehensive safety assessment.

Answer: Thank you for this suggestion. We think that it could be a good idea to include the techniques mentioned, given the differences found in transaminase levels between treated and untreated animals. However, we did not obtain liver and kidney samples from the animals used in this study, so we cannot provide results from histopathological evaluations. We also did not study biochemical profiles of the kidney. We will consider including these methods to improve our studies in the future. Furthermore, in the discussion we have highlighted that performing these methods could complete the study and we have mentioned the limitations of studying only enzyme levels and not including histopathological approaches.

  1. Discussion: Organize the discussion section to improve clarity and coherence. Address various aspects of the study one by one and compare your results with those of other relevant studies. Updating the references in the discussion section to include recently published papers will contribute to a more comprehensive and up-to-date discussion.

Answer: We have revised the discussion section and modified some aspects the reviewers suggested. We have tried to update the references.

  1. Conclusion: Include a concise conclusion to summarize the results of your study.

Answer: A brief conclusion section has been included at the end of the manuscript summarizing our results.

  1. Comments on the Quality of English Language: The manuscript can be checked for some minor English errors

Answer: The document has been reviewed by a native English speaker and we hope that errors have been corrected.

Reviewer 2 Report

Comments and Suggestions for Authors

In this manuscript, Gonçalves-Sánchez et. al evaluate the effectiveness and safety of eslicarbazepine acetate (ESL), a third-generation FDA approved antiepileptic drug (AED), in a model of audiogenic reflex seizures using the Genetic Audiogenic Seizures Hamster from Salamanca (GASH/Sal). Their research involved both acute and chronic administration of ESL in hamsters, assessing seizure severity using neuroethological methods, conducting behavioral tests, and monitoring hematological and biochemical profiles. The study concludes that both acute and chronic ESL administration effectively reduced seizure severity in the GASH/Sal model without significant difference in effectiveness, and no adverse effects were observed.

As the Authors mention in the manuscript, the effects of ESL have been tested in many animal models. While it is important to investigate the safety and efficacy of AEDs, it appears that audiogenic seizures represent a mostly unexplored niche. The Authors need to further elaborate on why this hamster model of audiogenic seizures is a particularly good addition to the field? From my understanding, audiogenic seizures are a rather small representation of global seizures in humans. It is noteworthy that this study potentially represents the first evaluation of ESL safety and efficacy in an animal model. However, the Authors are asked to clarify the work’s novelty and significance.

However, these points aside, the manuscript is solid and improved methods and animal models for seizure treatment and assessment are needed. Therefore, I recommend the manuscript be accepted only after the following specific concerns are addressed:

Specific comments to be addressed:

1. Further clarifications on the novelty and utility of this publication are needed. At the moment, it is not clear how relevant audiogenic seizures are to advancing human health. The Authors are encouraged to further elaborate.

2. While most of the experimental details were carefully written, the Authors did not explain the purpose of the initial 200 mg/kg dose of ESL before exposing the hamsters to acoustic stimulation. Please add more details.

3. The Authors focused most of their discussion on the effect of acute (150 mg/kg) and chronic (100 mg/kg) ESL. Can different treatment dose affect the results? For example, can you compare them at the same dose and/or more significantly different doses?

Author Response

Reviewer 2 

  1. Further clarifications on the novelty and utility of this publication are needed. At the moment, it is not clear how relevant audiogenic seizures are to advancing human health. The Authors are encouraged to further elaborate. The Authors need to further elaborate on why this hamster model of audiogenic seizures is a particularly good addition to the field? From my understanding, audiogenic seizures are a rather small representation of global seizures in humans. It is noteworthy that this study potentially represents the first evaluation of ESL safety and efficacy in an animal model. However, the Authors are asked to clarify the work’s novelty and significance.

Answer: In the introduction section we have added the aims of the study and why our study is important, as also suggested by the first reviewer. Furthermore, we have added what the particularities of animal models of audiogenic epilepsy are and their main advantages and how they can contribute to the advancement of human health.

  1. While most of the experimental details were carefully written, the Authors did not explain the purpose of the initial 200 mg/kg dose of ESL before exposing the hamsters to acoustic stimulation. Please add more details.

Answer: In the acute study, different doses (100, 150, 200 and 300 mg/kg) were evaluated to see which one had an anticonvulsant effect without showing serious side effects, since it was the first time it was evaluated in the GASH/Sal model. The 200 mg/kg dose was the first of the doses studied, after consulting the literature and discarding the 300 mg/kg because of serious side effects. We included those references in the discussion.

  1. The Authors focused most of their discussion on the effect of acute (150 mg/kg) and chronic (100 mg/kg) ESL. Can different treatment dose affect the results? For example, can you compare them at the same dose and/or more significantly different doses?

Answer: The acute study was carried out using doses of 100, 150 and 200 mg/kg, besides a dose of 300 mg/kg that was discarded because its side effects. In the chronic study the animals were treated with 100 mg/kg ESL, since all doses in the acute study had a similar anticonvulsant effect. Therefore, for both studies we have data of the effect of treatment with 100mg/kg ESL on seizures. However, it is true that for the open field tests, we evaluated different doses in both experiments. We have noted this limitation of the open field study for readers to keep in mind in the discussion section.

Reviewer 3 Report

Comments and Suggestions for Authors

Thank you for this interesting study. The paper is well presented and quite easy to read. I have some comments and suggestions to improve its quality. 

The introduction should include some development about the documented side-effects (biological and behavioral) of this drug and especially its bio-pharmacological common traits with carbamazepine. Since this research involves animals, it could make sense to mention the data available about carbamazepine and its use in pet species (dogs) as well as with the toxicology of this drug.

The Mat and Meths section: please move the lines 184-186 "Behavioral frequency was definbed ....observation period." to the Neuro-ethological paragraph, since it is clearly related to this part.

The authors study the biochemical profiles and especially the enzymatic possible modification, but they don't include a histological evaluation of the liver. One may suggest that there could be some hepatic lesions without consequence on the plasmatic levels of enzymes. Please justify or explain the absence of such analysis.

Except if I have missed something, the results for testing homoscedasticity to allow the use of ANOVA, are not cited. 

The authors use the same hamsters for repeated open field tests, it would be interesting to check if the repetition has an influence on the behavior. Repetition may lead to a decrease in the motivation to explore the arena and make that the behavior is significantly decreased.

The authors use the word "chronic", a term that may be discussed for such a short duration of treatment. If we compare with treatment in patients and the observation of biological disorders with the "sister molecule" carbamazepine, in human patients or in dogs, it is difficult to argue that we face a real "chronicity". According to this comment, I'd suggest to be more careful with stating the absence of toxicity. Again, the absence of discussion comparing this molecule with carbamazepine, is crucial. Including this discussion should lead to discuss the enzymatic induction by eslicarbazepine, that is not discussed in this paper, despite the use of the term "chronic". Could you justify it?

Author Response

Reviewer 3

  1. The introduction should include some development about the documented side-effects (biological and behavioral) of this drug and especially its bio-pharmacological common traits with carbamazepine. Since this research involves animals, it could make sense to mention the data available about carbamazepine and its use in pet species (dogs) as well as with the toxicology of this drug.

Answer: thank you for your comment. We have included these aspects in the introduction. Carbamazepine is not recommended for use in dogs because of a rapid induction of hepatic enzymes that eliminate the drug quickly.

  1. The Mat and Meths section: please move the lines 184-186 "Behavioral frequency was defined ....observation period." to the Neuro-ethological paragraph, since it is clearly related to this part.

Answer: We moved these lines following your advice.

  1. The authors study the biochemical profiles and especially the enzymatic possible modification, but they don't include a histological evaluation of the liver. One may suggest that there could be some hepatic lesions without consequence on the plasmatic levels of enzymes. Please justify or explain the absence of such analysis.

Answer: Your comment is accurate. Different publications point out that the pharmacokinetics of eslicarbazepine was not affected significantly due to mild to moderate liver failure. Since we did not collect histological samples from these animals, it is not possible to perform these techniques. However, in the discussion section we have reflected the limitations of our methodology.

  1. Except if I have missed something, the results for testing homoscedasticity to allow the use of ANOVA, are not cited. 

Answer: You are right, we tested homoscedasticity, but it was not cited in the text. This has now been corrected.

  1. The authors use the same hamsters for repeated open field tests, it would be interesting to check if the repetition has an influence on the behavior. Repetition may lead to a decrease in the motivation to explore the arena and make that the behavior is significantly decreased.

Answer: The effect of the repetition of the tests, without the effect of the drug, was carried out in animals in which only saline solution was administered. These are data from previous studies from our laboratory that are cited in the text (Cabral-Pereira et al., 2021). We described this in the discussion section.

  1. The authors use the word "chronic", a term that may be discussed for such a short duration of treatment. If we compare with treatment in patients and the observation of biological disorders with the "sister molecule" carbamazepine, in human patients or in dogs, it is difficult to argue that we face a real "chronicity". According to this comment, I'd suggest to be more careful with stating the absence of toxicity. Again, the absence of discussion comparing this molecule with carbamazepine, is crucial. Including this discussion should lead to discuss the enzymatic induction by eslicarbazepine, that is not discussed in this paper, despite the use of the term "chronic". Could you justify it?

Answer: we included a brief comparison between eslicarbazepine and carbamazepine in the introduction. In the discussion, we mention the differences in the metabolism of ESL between humans and other species, including the hamster (Bialer & Soares da Silva., 2012). We used the term chronic since for other research in animal models, similar periods of time (two weeks or 15 days) are considered chronic (Swinyard et al., 1987).

Round 2

Reviewer 3 Report

Comments and Suggestions for Authors

Thank you for editing and improving your paper. I am fine with this new version. I have just an advice, or a comment, related to the use of the word "chronicity", when there are just a maximum of 2 weeks. This is quite a short treatmen,t, to be considered as chronic, when animals receiving this treatment in a chronic way (to treat epilepsy) are administered for months chan not for years. I still recommend to avoid to use this wording, that may suggest that you support to consider this drug as absolutely safe.

Author Response

Dear reviewer

Thank you for your thoughtful comments.

Following your advice, we have modified the term "chronic" for “subchronic”.

Best regards.

Dolores E. López